# Parental and Pediatricians’ Attitudes towards COVID-19 Vaccination for Children: Results from Nationwide Samples in Greece

**DOI:** 10.3390/children9081211

**Published:** 2022-08-11

**Authors:** Evangelia Steletou, Theodoros Giannouchos, Ageliki Karatza, Xenophon Sinopidis, Aggeliki Vervenioti, Kyriakos Souliotis, Gabriel Dimitriou, Despoina Gkentzi

**Affiliations:** 1Department of Pediatrics, Patras Medical School, 26504 Rio, Greece; 2Department of Health Services Policy and Management, Arnold School of Public Health, University of South Carolina, Columbia, SC 29208, USA; 3Department of Social and Education Policy, University of Peloponnese, 20131 Corinth, Greece; 4Health Policy Institute, 15123 Maroussi, Greece

**Keywords:** COVID-19, vaccination, parents, pediatricians, children, attitudes, intention

## Abstract

Although many studies have examined factors associated with COVID-19 vaccination and healthcare professionals’ attitudes towards vaccines, less is known about parents’ and pediatricians’ attitudes towards COVID-19 vaccination for children. Using two cross-sectional surveys from November to December 2021 in Greece, we aimed to assess parental intention to vaccinate their 5 to 17 years old children against COVID-19 and to evaluate pediatricians’ attitudes towards children’s vaccination against COVID-19. Overall, 439 parents and 135 pediatricians participated. Of them, 240 (54.7%) intended to vaccinate their children against COVID-19. The most commonly reported reasons for non-intention were the short length of clinical trials and the fear of side effects. Only 16.6% of non-intenders would vaccinate their children if the pediatrician recommended it. The factors associated with higher intention to get vaccinated were a parent’s own vaccination against COVID-19, trust in official healthcare guidelines, increased trust in the state and the healthcare system during the pandemic, and older age. Of the pediatricians, 92.6% recommended children’s vaccination, and 75.6% agreed with mandating the vaccine. These findings suggest the need to tailor easy-to-understand messages by well-informed pediatricians to address safety concerns, educate, and clarify misconceptions through targeted interventions among those who currently do not wish to vaccinate their children.

## 1. Introduction

Individuals with a chronic disease, both children and adults, are at a higher risk for severe Coronavirus Disease 2019 (COVID-19) disease and complications from it [1]. Although children are, in general, mildly affected by COVID-19 compared to adults, they might suffer from a rare yet serious condition after contracting COVID-19, the Multisystem Inflammatory Syndrome in Children (MIS-C), with life-threatening complications in some cases [2]. Therefore, prevention efforts, including vaccination, are critical to gradually restore children’s school and social life and to further contribute to reaching herd immunity thresholds. Due to the emergence of new variants, COVID-19-associated hospitalization of children and adolescents presented a rapid increase at the end of 2021, especially among children between 0 to 4 years of age who were not considered eligible for vaccination [3]. However, hospitalization rates were lower in vaccinated adolescents, who became eligible for vaccination during that period compared to their unvaccinated counterparts [3].

Although children aged 5 years and older are eligible to receive a COVID-19 vaccine, vaccination rates are lower among children compared to adults [4]. Despite the rapidly evolving evidence of COVID-19 vaccine efficacy, only 30 to 63 percent of parents and caregivers were willing to vaccinate their children against the disease [5,6]. Apart from COVID-19 vaccines, a general decline in outpatient pediatric visits and routine vaccinations has also been documented during the pandemic [7].

Vaccine hesitancy, defined as vaccine reluctance or refusal despite vaccine availability, has already been characterized by the World Health Organization (WHO) as one of the major threats to public health, even before the COVID-19 pandemic [8]. Multiple factors have already been associated with vaccine hesitancy among adults for both themselves and their children, such as socioeconomic factors and health status, personal beliefs, social media influence, perceived childhood COVID-19 susceptibility and severity, vaccine safety and efficacy concerns, and physicians’ recommendations and beliefs [5,9,10,11]. For children in particular, primary care providers’ and pediatricians’ recommendations play a critical role to foster an environment of trust for the widespread and targeted roll-out of COVID-19 vaccines [12,13]. Despite the existing evidence on factors and characteristics associated with vaccination uptake, less is known about both parental and pediatricians’ attitudes towards COVID-19 vaccination for children from 5 to 17 years of age in Greece, which was authorized at the end of 2021 [14].

The aims of this study were (1) to examine parental intentions to vaccinate their 5- to 17-year-old children for COVID-19, (2) to estimate parental characteristics, factors, attitudes, and beliefs associated with higher vaccination intention for their children, and (3) to explore pediatricians’ attitudes towards vaccination of children 5 to 17 years old in Greece. Understanding the factors that influence the parental decision to vaccinate their offspring against the COVID-19, as well as pediatricians’ current opinions, is crucial to developing targeted policies aimed to increase the COVID-19 vaccination uptake among this population.

## 2. Materials and Methods

### 2.1. Study Design, Sampling, and Questionnaire for Parents

An anonymous questionnaire was administered to a nationwide sample of parents and caregivers of children aged between 5 and 17 years in November and December of 2021. The vaccine was under approval in Greece during the study period and was eventually recommended by the Hellenic Immunization Committee at the end of December 2021. The questionnaire was developed by two members of the research team (ES, DG) via a literature review [5,6,13,15]. Prior to study initiation, the questionnaire was piloted in a sample of twenty parents, and appropriate adjustments were made.

Phone interviews were conducted with parents of one or more children using structured questionnaires. The study sample was randomly selected from the national phone-number registry using a random stratified selection process. Telephone numbers of businesses or public services were excluded. Inclusion criteria were fluency in Greek language, being 17 years old or older, and having a child between the age of 5 to 17 years. Responders who stated that they did not have any children were excluded from participation.

The phone calls were carried out by the research team with the aid of a market research company with previous experience in demographic surveys. An overview of the study objectives was provided to potential participants, and verbal consent was obtained. The questionnaire was administered in Greek and included 20 questions divided into three parts. The first part (eight questions) collected sociodemographic, contextual, and clinical information. The second part (eight questions) assessed parental and other family members’ COVID-19 vaccine uptake, intention to vaccinate their child, whether they know about approval of COVID-19 vaccines for children, whether their pediatrician’s recommendation would influence their decision, reasons for not intending to vaccinate their offspring as well as sources of information on vaccines. In addition, information about children’s flu vaccine uptake during this season was collected. The third part (four questions) evaluated parents’ opinions about mandating COVID-19 vaccines for both adults and children, how much they trust official healthcare guidelines, and whether their trust in the state and healthcare system changed during the pandemic or not.

### 2.2. Study Design, Sampling, and Questionnaire for Pediatricians

Electronically structured questionnaires for pediatricians were similarly developed via a literature review and were distributed to all registered practicing pediatricians (~3700) via their personal email, obtained from the registry of the Greek Medical Association. The questionnaire included 19 questions which were divided into 3 parts. The first part (12 questions) collected sociodemographic, contextual, and clinical information. The second part (three questions) evaluated pediatricians’ own COVID-19 vaccine uptake, whether pediatricians recommend parents to vaccinate their children against COVID-19, and the reasons why they might not recommend it. The last part (four questions) assessed pediatricians’ attitudes about mandating COVID-19 vaccines for adults and children, how much they trust official healthcare guidelines regarding COVID-19, and whether their trust in the authorities and healthcare system changed during the pandemic.

### 2.3. Ethical Approval

The study was conducted according to the guidelines of the Declaration of Helsinki and approved by the Research Ethics Committee of the University of Patras, Greece (ID: 8135/01.09.21).

### 2.4. Statistical Analyses

Descriptive analyses were initially conducted for parents stratified by their intention to vaccinate their children against COVID-19 across all sociodemographic and contextual characteristics and factors, as well as COVID-19-related attitudes and vaccination-related information, and their trust in the healthcare system and guidelines. Since all questions were dichotomous or categorical, two-tailed Pearson’s χ2 tests were used to examine statistical significance. Multivariable logistic regression was used to estimate the association between parental COVID-19 vaccination intention for their children and all characteristics and factors included in the descriptive analyses. Standard errors were clustered at the area of residence to control for unobserved, time-invariant, regional differences. We used 95% confidence intervals, and statistical significance was considered at *p* < 0.05. Pediatricians’ questionnaires were also descriptively analyzed to identify their attitudes, opinions, and recommendations toward COVID-19 vaccination of children from 5 to 17 years old. Data were collected in Microsoft Excel and all statistical analyses were conducted using Stata version 17.0 (StataCorp, College Station, TX, USA).

## 3. Results

### 3.1. Descriptive Characteristics of Parents

Out of 555 eligible parents randomly approached over the phone, 439 parents fully completed the survey and were included in the analyses (response rate: 88.2%). The majority were females (58.8%), aged 40 to 54 years (68.8%), married (89.7%), fully employed (77.2%), and residing in urban areas (83.4%) (Table 1). Most had one (47.2%) or two underage children (41.2%), and 80.2% reported that their children’s health status was very good. Almost three in every four parents (74.5%) indicated that they were vaccinated against COVID-19, and the majority (93.2%) knew that the vaccine had been approved for children. In total, 62% were willing to vaccinate their children against COVID-19 if the pediatricians recommended it, while only 23.9% agreed with mandating the vaccine for children and 41.9% for adults. Most parents relied on their pediatrician (28%) or other healthcare professionals (39.6%) to decide whether to vaccinate their children against COVID-19.

### 3.2. Parental Intention to Vaccinate Their Children of 5 to 17 Years of Age

Almost all parents who intended to vaccine their children were aware of the vaccine’s approval for children and would vaccinate their children if the pediatrician recommended it, while both shares were lower among non-intenders (approval knowledge: 96.7% versus 88.9%, *p* = 0.001; vaccinate if pediatrician recommended: 99.6% versus 16.6%, *p* < 0.001) (Table 2). Attitudinal differences between intenders and non-intenders were also observed (Table 2). Parents who intended to vaccinate their children against COVID-19 exhibited disproportionately higher reliance on pediatricians (40.0% versus 13.6%, *p* < 0.001) or other healthcare professionals (48.8% versus 28.6%, *p* < 0.001) as their main sources to decide about their children’s vaccination, and they also reported higher support for mandating vaccination both for adults (64.2% versus 15.1%, *p* < 0.001) and children (43.7% versus 0.0%, *p* < 0.001). Similarly, most intenders reported a lot or very much trust in official healthcare guidelines compared to non-intenders (84.6% versus 29.1%, *p* < 0.001) and an increase in their trust in the state and the healthcare system during the pandemic relative to non-intenders (27.1% versus 2.5%. *p* < 0.001). Flu vaccination was also higher in children of parents who intended to vaccinate them against COVID-19 compared to those who did not (31.3% versus 14.6%, *p* < 0.001).

Among the 199 (45.3%) parents who did not intend to vaccinate their children against COVID-19, the most commonly reported main reasons were the short length of clinical trials (45.2%) and fear of side-effects (24.1%) (Table 3). About one in every ten non-intenders responded that the primary reason for not intending to vaccinate their children was related to the fact that children are not in danger of COVID-19-related adverse outcomes, while another 8.0% doubted the effectiveness of vaccines. Only 2.0% reported non-pediatrician recommendation as the main reason against COVID-19 vaccination.

Table 4 presents the multivariable logistic regression estimates of parental characteristics and attitudes associated with their intention to vaccinate their children against COVID-19. The strongest factors associated with vaccination intention were the parent’s own vaccination against COVID-19 (Adjusted Odds Ratio (aOR) = 9.14, 95% Confidence Intervals (CI) = 4.18–20.01, *p* < 0.001), trusting official healthcare guidelines very much compared to no trust (aOR = 11.00, 95% CI = 4.78–25.28, *p* < 0.001), increased trust in the state and the healthcare system during the pandemic compared to decreased trust (aOR = 9.05, 95% CI = 4.36–18.78, *p* < 0.001), and being older compared to those who were 25 to 39 years of age (40–54: aOR = 11.00, 95% CI = 4.59–26.36, *p* < 0.001; 55 to 64: aOR = 11.46, 95% CI = 3.25–40.36, *p* < 0.001). Parents who agreed with mandating COVID-19 vaccination for all children and those who had vaccinated their children against the flu were also more likely to intend to vaccinate their children against COVID-19 compared to those who did not support the mandate and those who had not vaccinated their children against the flu, respectively (mandate: aOR = 2.78, 95% CI = 1.22–6.35, *p* = 0.015; flu vaccination: aOR = 3.73, 95% CI = 1.34–10.41, *p* = 0.012). In contrast, compared to parents who mostly relied on pediatricians to decide about their children’s vaccination against COVID-19, those who relied on personal beliefs or other sources (TV, internet, social media) were less likely to intend to vaccinate their children against COVID-19 (personal views/beliefs: aOR = 0.37, 95% CI = 0.17–0.81, *p* = 0.013; other sources: aOR = 0.09, 95% CI = 0.02–0.12, *p* < 0.001).

### 3.3. Pediatricians’ Characteristics, Attitudes, Beliefs, and Recommendations for COVID-19 Vaccination of Children 5 to 17 Years of Age

Overall, 135 pediatricians completed the questionnaire (response rate 4%). Pediatricians’ characteristics, attitudes, beliefs, and recommendations for COVID-19 vaccination are presented in Table 5 and Table 6. On average, they were 50.9 years of age (standard deviation (SD) = 10.1), with 18.5 years of practice (SD = 10.8), and who vaccinated around 102.1 (SD = 10.2.4) children per year. Overall, 92.6% of pediatricians recommended children’s vaccination against COVID-19, and 47.4% agreed with mandating the vaccine for children. Only two pediatricians (1.5%) did not suggest COVID-19 vaccination for all children, while eight pediatricians (5.9%) suggested COVID-19 only for high-risk children. The main reason against vaccination for all children was the short duration of clinical trials and safety concerns, which was reported by 8 of these 10 pediatricians. Finally, only 1.5% and 3.7% of pediatricians reported no or limited (a bit) trust in experts’ guidelines and recommendations on COVID-19.

## 4. Discussion

This study investigated parental intention to vaccinate their children against COVID-19 at the end of 2021, when approval for childhood vaccination of children aged 5–11 years COVID-19 was still pending in Greece. According to our results, the percentage of Greek parents who intended to vaccinate their children from 5 to 17 years of age was 54.7%, while 92.6% of pediatricians recommended pediatric vaccination against COVID-19. These findings are consistent with global and country-specific estimates during the same period [13,16,17,18,19,20,21].

Among the factors which were significantly associated with parents’ positive attitude towards children’s vaccination against COVID-19 were pediatricians’ and other healthcare professionals’ recommendations and parents’ own vaccination status. The latter has been noted in studies conducted worldwide, where childhood COVID-19 vaccine acceptance was strongly associated with parents’ intentions to get vaccinated themselves [19,20,21]. In contrast, reliance on personal beliefs and other non-healthcare provider resources were inversely associated with the parental intention to vaccinate their children against COVID-19. Interestingly, most parents who were not willing to vaccinate their child insisted on their decision and reported that they would not be persuaded to vaccinate their child even if their pediatrician recommended it. These results are in line with a recent study on the topic, which found that some parents do not trust healthcare providers, particularly when they believe that they are not well informed about the vaccine or its ingredients [22].

Both in our study and other studies among parents, the primary reasons for refusing COVID-19 vaccination were consistent and mostly related to vaccine safety concerns [3,19,20,21]. Parents’ opinion on the subject is opposed to national hospital surveillance data in the United States from July 2021 until January 2022, which revealed that hospitalization rates of COVID-19 among children and adolescents increased rapidly during the last 2 weeks of December 2021, coinciding with the Omicron variant, particularly among those aged 0–4 years, who were not yet eligible for vaccination [3]. During Delta and Omicron predominance periods, rates of hospitalization were lower among fully vaccinated compared to unvaccinated adolescents [3]. Therefore, strategies to convince parents that children are also at increased risk of severe COVID-19 and hospitalization are critical to increase vaccination rates and prevent severe illness [1,3].

Given the relationship of trust between Greek parents and pediatricians, which develops during childhood because of routine visits, pediatricians play a critical role in convincing parents to vaccinate their children against COVID-19. As our data indicate, almost all pediatricians in Greece suggest children’s COVID-19 vaccination. However, the finding regarding the limited uptake of vaccination among parents who did not initially intend to vaccinate their children even after the pediatricians’ recommendation raises concerns regarding communication barriers between parents and providers. As such, beyond educational initiatives tailored to parents who rely on less trustworthy resources to decide whether to vaccinate their children, efforts are needed to promote more efficient communication between pediatricians and parents. Easy-to-understand messages by well-informed pediatricians might address concerns, educate, and clarify misconceptions through targeted language across vaccine hesitant individuals. Apart from pediatricians, consistent public health guidelines and recommendations without confusing messages are critical to enhance and bolster trust in authorities, which, based on our and previous findings, are pivotal to achieve higher vaccination rates [20,21,23,24].

This study may differ from other similar studies, since our sample was randomly selected from the national phone number registry. It includes parents from all the regions of Greece, which is one of the study’s major strengths. Our results might be generalizable, with caution, to other countries in the region or within Europe with similar cultural and healthcare system-related factors. We acknowledge though that our study, as a pragmatic one, is not without limitations. First of all, the participation in our study was voluntary, therefore, vaccine-hesitant parents or pediatricians may have deliberately decided not to participate, causing selection bias. In addition, it is worth mentioning the bias resulting from the fact that only households with a landline phone and that were in the national phone registry were considered. Furthermore, in the present study, we only point out the intention to vaccinate, and this is not necessarily translated as the actual vaccine uptake. Consequently, taking into account the dynamic nature of the pandemic, we cannot be certain of what parents will eventually do. Moreover, although around 4% of all pediatricians in Greece completed the survey, participating pediatricians may have answered the questions in a way such that they can be viewed favorably by others, resulting in social-desirability bias. Therefore, pediatricians’ hesitancy for the vaccination against COVID-19 might also be under-reported.

## 5. Conclusions

Although the majority of pediatricians are in favor of COVID-19 vaccine for children 5 to 17 years of age and they suggest vaccination to parents, about 45% of Greek parents do not intend to vaccinate their children against COVID-19 despite the recommendation. Our results highlight the need to promote more efficient communication between pediatricians and parents through easy-to-understand messages and, overall, to enhance public trust in the healthcare system. This could, in turn, contribute to the development of targeted strategies to address concerns regarding the COVID-19 vaccine for children in order to increase vaccination uptake.

## Figures and Tables

**Table 1 children-09-01211-t001:** Descriptive characteristics overall and stratified by parental intention to vaccinate their children.

Intention to Vaccinate Children Against COVID-19				
	All	No	Yes	*p*-Value
**N (%)**	439 (100)	199 (45.3)	240 (54.7)	0.335
**Gender (%)**				
Male	41.2	43.7	39.2	
Female	58.8	56.3	60.8	<0.001
**Age groups (%)**				
25 to 39	22.8	32.2	15.0	
40 to 54	68.8	61.8	74.6	
55 to 64	8.4	6.0	10.4	0.411
**Marital status (%)**				
Married	89.7	88.4	90.8	
Single/Divorced/Widowed	10.3	11.6	9.2	0.001
**Education (%)**				
High School or less	39.6	49.2	31.7	
Bachelors	39.7	32.7	45.4	
Masters/PhD	20.7	18.1	22.9	0.286
**Unemployed (%)**				
No	77.2	74.9	79.2	
Yes	22.8	25.1	20.8	0.779
**Area of residence (%)**				
Urban	83.4	83.9	82.9	
Rural	16.6	16.1	17.1	0.039
**Number of underage children (%)**				
One	47.2	40.7	52.5	
Two	41.2	45.2	37.9	
Three or more	11.6	14.1	9.6	0.120
**Children’s health status (%)**				
Very good	80.2	79.9	80.4	
Good	9.8	7.5	11.7	
Average or lower	10.0	12.6	7.9	<0.001
**Parent is vaccinated against COVID-19 (%)**				
No	25.5	54.3	1.7	
Yes	74.5	45.7	98.3	<0.001
**Other family members is vaccinated against COVID-19 (%)**				
No	24.4	44.7	7.5	
Yes	75.6	55.3	92.5	0.001

**Table 2 children-09-01211-t002:** Knowledge and attitudes of parents on COVID-19 vaccination overall and stratified by their intention to vaccinate their children.

Intention to Vaccinate Children Against COVID-19				
		Yes	No	
**Do you know that COVID-19 vaccination has been approved for children aappapproved** **for children? (%)**				
Yes	93.2	88.9	96.7	
No	6.8	11.1	3.3	<0.001
**Would you vaccinate your children against COVID-19 if recommended by the pediatrician? (%)**				
No	34.6	76.4	0.0	
Yes	62.0	16.6	99.6	
Not sure	3.4	7.0	0.4	<0.001
**What sources do you mostly rely on to decide about your children’s vaccination against COVID-19? (%)**				
Pediatrician	28.0	13.6	40.0	
Other healthcare professionals	39.6	28.6	48.8	
Personal views/beliefs	15.5	25.6	7.1	
Other sources (TV, Internet, Social)	8.4	14.1	3.8	
Do not wish to answer	8.4	18.1	0.3	<0.001
**Did your children receive the flu vaccine this year? (%)**				
No	76.3	85.4	68.7	
Yes	23.7	14.6	31.3	<0.001
**Do you agree with mandating vaccination for adults? (%)**				
No	42.1	75.9	14.2	
Yes	41.9	15.1	64.2	
Yes, but for certain groups only	16.0	9.0	21.7	<0.001
**Do you agree with mandating vaccination for children? (%)**				
No	76.1	100.0	56.3	
Yes	23.9	0.0	43.7	<0.001
**How did your trust in the state and healthcare system change during the COVID-19 pandemic? (%)**				
Decreased	52.4	75.9	32.9	
No change	31.7	21.6	40.0	
Increased	15.9	2.5	27.1	<0.001
**How much do you trust official healthcare guidelines? (%)**				
No trust	22.1	41.2	6.2	
A bit	18.4	29.7	9.2	
A lot	35.8	26.1	43.8	
Very much	23.7	3.0	40.8	<0.001

**Table 3 children-09-01211-t003:** Primary reason for not intending to vaccinate children against COVID-19 (N = 199).

Reasons	N (%)
Short length of clinical trials	90 (45.2)
Fear for side-effects	48 (24.1)
Children are not in danger of adverse COVID-19 related outcomes	20 (10.1)
Doubt about vaccine effectiveness	16 (8)
Vaccination serves other purposes	10 (5)
Not recommended by the pediatrician	4 (2)
Child already had COVID-19	2 (1)
Other reasons (i.e., religion)	9 (4.6)

**Table 4 children-09-01211-t004:** Multivariate logistic regression estimates of the association between parental intention to vaccinate their children against COVID-19 and parental characteristics, opinions, and preferences.

	aOR	95% CI	*p*-Value
**Gender (Ref. Male)**			
Female	1.24	0.40–3.89	0.370
**Age groups (Ref. 25 to 39)**			
40 to 54	11.00	4.59–26.36	<0.001
55 to 64	11.46	3.25–40.36	<0.001
**Marital status (Ref. Married)**			
Single/Divorced/Widowed	0.66	0.34–1.27	0.213
**Education (Ref. High School or less)**			
Bachelors	2.24	1.07–4.72	0.033
Masters/PhD	1.60	0.71–3.60	0.260
**Unemployed (Ref. No)**			
Yes	0.71	0.27–1.85	0.486
**Area of residence (Ref. Urban)**			
Rural	1.78	0.53–5.92	0.350
**Number of children (Ref. One)**			
Two	0.47	0.15–1.43	0.183
Three or more	1.44	0.35–5.91	0.614
**Children’s health status (Ref. Very good)**			
Good	2.63	1.05–6.55	0.038
Average or lower	1.12	0.36–3.50	0.851
**Parent vaccinated against COVID-19 (Ref. No)**			
Yes	9.14	4.18–20.01	<0.001
**Other family member is vaccinated against COVID-19 (Ref. No)**			
Yes	2.15	0.34–13.73	0.419
**What sources do you mostly rely on to decide about your childrens’ vaccination against COVID-19? (Ref. Pediatrician)**			
Other healthcare professionals	1.03	0.50–2.12	0.938
Personal views/beliefs	0.37	0.17–0.81	0.013
Other sources (TV, Internet, Social)	0.09	0.02–0.35	<0.001
Do not wish to answer	0.02	0.00–0.12	<0.001
**Do you agree with mandating vaccination for adults? (Ref. No)**			
Yes	2.78	1.22–6.35	0.015
Yes, but for certain groups only	3.86	0.82–18.19	0.088
**Did your children receive the flu vaccine this year? (Ref. No)**			
Yes	3.73	1.34–10.41	0.012
**How did your trust in the state and healthcare system change** **during the COVID-19 pandemic? (Ref. Decreased)**			
No change	3.93	1.75–8.79	0.001
Increased	9.05	4.36–18.78	<0.001
**How much do you trust official healthcare guidelines? (Ref. No trust)**			
A bit	0.71	0.32–1.56	0.399
A lot	1.10	0.39–3.10	0.856
Very much	11.00	4.78–25.28	<0.001

**Table 5 children-09-01211-t005:** Pediatricians’ sociodemographic characteristics.

N	135	
**Gender**		
Male	48	35.6%
Female	87	64.4%
**Age**		
Average (SD)	50.9 (10.1)	
Categories		
32–41	26	19.3%
42–51	48	35.5%
52–61	39	28.9%
62+	22	16.3%
**Years of practice**		
Average (SD)	18.5 (10.8)	
Categories		
1 to 9	30	22.2%
10 to 19	42	31.1%
20 to 29	42	31.1%
30 or more	21	16.6%
**Healthcare sector of employment**		
Public	42	31.1%
Private	92	68.2%
Both	1	0.7%
**How many children do you vaccinate each year?**		
Average (SD)	102.1 (102.4)	
Categories		
0	7	5.2%
1 to 49	39	28.8%
50 to 99	29	21.5%
100 to 149	26	19.3%
150 or more	34	25.2%
**Area of practice**		
Urban	110	81.5%
Rural	25	19.5%
**Marital status**		
Married/cohabitating	106	78.5%
Single	20	14.8%
Other (divorced, widowed)	9	6.7%
**Self-reported health status**		
Average	8	5.9%
Good	47	34.8%
Very good	80	59.3%
**Do you have underaged children?**		
No	78	57.8%
Yes	57	42.2%
**Do you belong in a high-risk group for COVID-19 due to a chronic condition?**		
No	113	83.7%
Yes	22	16.3%

**Table 6 children-09-01211-t006:** Pediatricians’ attitudes, beliefs, and recommendations.

N	135	
**Have you been diagnosed with COVID-19?**		
No	115	85.2%
Yes	20	14.8%
**Has anyone in your family been diagnosed with COVID-19?**		
No	86	63.7%
Yes	49	36.3%
**Have you been vaccinated against COVID-19?**		
No	0	0.0%
Yes	135	100.0%
**Do you suggest to parents to vaccinate their children against COVID-19?**		
No	2	1.5%
Yes, but only for high-risk children	8	5.9%
Yes, for all children	125	92.6%
**What are the reasons for which you don’t suggest vaccination to all children for COVID-19? (n = 10)**		
Children are not in danger	1	
Short duration of clinical trials/safety concerns	8	
Legal concerns in case of adverse effects	1	
**Do you agree with mandating COVID-19 vaccination for adults?**		
No	10	7.4%
Yes, but only for certain populations	23	17.0%
Yes	102	75.6%
**Do you agree with mandating COVID-19 vaccination for children?**		
No	71	52.6%
Yes	64	47.4%
**Do you trust the experts’ guidelines and recommendations on COVID-19?**		
Not at all	2	1.5%
A bit	5	3.7%
Enough	67	49.6%
A lot	61	45.2%
**How did your trust in the authorities and healthcare system change during the pandemic?**		
Decreased	44	32.6%
No change	61	45.2%
Increased	30	22.2%

## Data Availability

Additional study data are available on request.

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
