# Peer review of "Parental and Pediatricians’ Attitudes towards COVID-19 Vaccination for Children: Results from Nationwide Samples in Greece"

_children, 2022, doi:10.3390/children9081211_

Round 1

Reviewer 1 Report

This article investigates parents' intention to vaccinate their children against Covid-19 and pediatricians' recommendation in Greece end of 2021, and the relation between the two and other factors. It competes the number of studies on the subject in other countries, and shows similar patterns.

Table 1: 

- labeling: "all" under the label of intention to vaccinate. Maybe rename % of subjects

- It would make more sense to change and have 100% sum on the horizontal row rather than vertical column, eg., % of intention to vaccinate among urban population, since the different sizes of population make it difficult to sense the difference (same for text)

Discussion:

Selection bias parents: 88%return rate is quite high, therefore less bias than for pediatricians. But it's worth it mentioning the bias resulting from the fact that only households with a landline phoneline (and being in the national phone register -compulsory?) were considered. https://data.worldbank.org/indicator/IT.MLT.MAIN.P2?locations=GR

"Our results highlight the need to promote more efficient communication between pedia- 295 tricians and parents through easy-to-understand messages and could contribute to the 296 development of targeted strategies to address concerns regarding COVID-19 vaccine for 297 children in order to increase vaccination rates" is in my opinion in contradiction with what you state in the discussion, namely that "Interestingly, most parents who were not willing to vaccinate their child insisted on their 243 decision and reported that they would not be persuaded to vaccinate their child even if 244 their pediatrician recommended it". This is also my experience, that people not trusting HCsytem won't be convinced by any form of message by HC professionals (it might rather make the aversion stronger). This means a main objective would be on the long term to instore means to restore trust in the system (incl to the independence of research and to the pharmaceutical industry).

Author Response

We would like to thank the reviewer for the positive feedback on our manuscript

Table 1: labeling: "all" under the label of intention to vaccinate. Maybe rename % of subjects It would make more sense to change and have 100% sum on the horizontal row rather than vertical column, eg., % of intention to vaccinate among urban population, since the different sizes of population make it difficult to sense the difference (same for text)

Thank you for the comment. Following your advice we have amended  Table 1 adding the clarification % after each variable tested to make this more clear to the readers.

Discussion:

Selection bias parents: 88%return rate is quite high, therefore less bias than for pediatricians. But it's worth it mentioning the bias resulting from the fact that only households with a landline phoneline (and being in the national phone register -compulsory?) were considered. https://data.worldbank.org/indicator/IT.MLT.MAIN.P2?locations=GR

This is indeed a very useful comment which we have now added to the limitations of our study in the revised version of the manuscript as follows:  In addition, it's worth it mentioning the bias resulting from the fact that only households with a landline phoneline and being in the national phone registry were considered.

"Our results highlight the need to promote more efficient communication between pedia- 295 tricians and parents through easy-to-understand messages and could contribute to the 296 development of targeted strategies to address concerns regarding COVID-19 vaccine for 297 children in order to increase vaccination rates" is in my opinion in contradiction with what you state in the discussion, namely that "Interestingly, most parents who were not willing to vaccinate their child insisted on their 243 decision and reported that they would not be persuaded to vaccinate their child even if 244 their pediatrician recommended it". This is also my experience, that people not trusting HCsytem won't be convinced by any form of message by HC professionals (it might rather make the aversion stronger). This means a main objective would be on the long term to instore means to restore trust in the system (incl to the independence of research and to the pharmaceutical industry).

Thank you for the useful comment. Following your advice we have changed the conclusion incorporating your suggestion in the revised version of the manuscript as follows:

Our results highlight the need to promote more efficient communication between pediatricians and parents through easy-to-understand messages and overall to enhance public trust in the healthcare system. This could in turn contribute to the development of targeted strategies to address concerns regarding COVID-19 vaccine for children in order to increase vaccination uptake.

Reviewer 2 Report

Many thanks to the authors for carrying out the research and presenting the results in the article, which I had the pleasure of reviewing. 

The article is very interesting and interesting. 

I have a few comments/proposals that I think are worth including in the article to improve its methodological design.

1. In the Materials and Methods chapter, I suggest introducing additional subsections:

Study Design

Sample Selection 

Questionnaires

The sections on parents and pediatricians should be very clearly separated in each of these subsections. 

2. Line 84 and on: was the study conducted by a research company? If so, it is worth describing this and adding information on the methodological assumptions of the survey. 

3. The Statistical Analysis (line 118) lacks reference to the statistical error for each group. 

4. Line 150, Table 1., age: Was it a decision by the researchers that there were no subjects younger than 25 (one imagines there could have been) years old and older than 64?

5. Breakdown of Table 1 From question: Do you know that the COVID-19 vaccination... I propose to create a separate table and refer briefly descriptively to its results.

6. Subdivision 3.2 lines 152-159 and onwards refers to what is included in the first part of Table 1.  - according to the principle of writing this type of article, there is no reason to repeat it descriptively. 

7. I propose to use lines 164-175 as a description for the table that will be created after splitting Table 1 into two, as suggested above. I also suggest shortening this description a bit to avoid repeating everything included in the table. 

8. Table 2, line 186, has unclear designations. It is not clear whether N=199 is the sum of the following numbers. The designation n=199 should be in the table's title, not in the first line. In the table, in addition to percentages, there should be numbers. 

9. In subsection 3.3, line 212, I propose to separate, as in the case of parents: Table 4 into two: characteristics of the respondents and the second table from the question: Have you been diagnosed...

10. Please consider bolding the questions in the tables - this will make distinguishing the question from the answer easier. 

11. The results regarding pediatricians are treated very gloomily, perhaps because this is a small group; nevertheless, it would be possible to see, for example, how pediatricians who have children and pediatricians who do not have children relate to some of the questions and how those who agree with approving vaccination for children and those who disagree with it respond. 

12. The discussion needs to be improved by referring to more literature and comparing other studies. Many interesting papers have been published on parents and COVID-19 vaccination and pediatricians and COVID-19 vaccination. 

13. Should the paragraph from line 278 be considered a Limitation of the Study? If so, it should be noted. 

14. I would also suggest highlighting the strengths of the study. E.g., I disagree with the sentence lines:278-281 are strengths, and in my opinion, one might be tempted, with some caution, to try to transfer the results to countries in the region or the EU. 

15. One last technical note: Author Contributions, Informed Consent Statement, and Conflicts of Interest without quotation marks.

Author Response

We would like to thank the reviewer for the positive feedback on our manuscript and for giving us the opportunity to improve our manuscript

  1. In the Materials and Methods chapter, I suggest introducing additional subsections:

Study Design Sample Selection  Questionnaires

The sections on parents and pediatricians should be very clearly separated in each of these subsections. 

Thank you for the comment following which we have now added the additional subsections in the revised manuscript version

Materials and Methods

2.1. Study design, sampling and questionnaire for parents

2.2. Study design, sampling and questionnaire for pediatricians

2.3. Ethical approval

2.4. Statistical Analyses

  1. Line 84 and on: was the study conducted by a research company? If so, it is worth describing this and adding information on the methodological assumptions of the survey. 

As stated in the Methods Section of the manuscript our  survey was carried out with the aid of  a market research company with expertise in demographic surveys, with guidance from the authors. The study sample was selected at random from national phone-number databanks using a random stratified selection process from the national landline telephone directory. Telephone numbers belonging to individuals only were included and numbers of businesses or public services were excluded.  Inclusion criteria were fluency in Greek language, being 17 years old or older, and having a child between the ages of 5 to 17 years. If the responders stated that they did not have any children, they were excluded from the study.

As the sample was selected at random from the national phone registry we have added the following comment in the limitation section of the revised section of the manuscript: In addition, it's worth it mentioning the bias resulting from the fact that only households with a landline phoneline and being in the national phone registry were considered.

  1. The Statistical Analysis (line 118) lacks reference to the statistical error for each group. 

We have now added the following sentence in the revised manuscript version: We used 95% confidence intervals and statistical significance was considered at p<0.05

  1. Line 150, Table 1. age: Was it a decision by the researchers that there were no subjects younger than 25 (one imagines there could have been) years old and older than 64?

We did not have any  parents younger than 25 or older than 64 in our cohort and hence we did not include such age groups in the analyses.

  1. Breakdown of Table 1 From question: Do you know that the COVID-19 vaccination... I propose to create a separate table and refer briefly descriptively to its results.

  1. Subdivision 3.2 lines 152-159 and onwards refers to what is included in the first part of Table 1.  - according to the principle of writing this type of article, there is no reason to repeat it descriptively. 

  1. I propose to use lines 164-175 as a description for the table that will be created after splitting Table 1 into two, as suggested above. I also suggest shortening this description a bit to avoid repeating everything included in the table. 

Following you comments 5,6,7 we have broken down Table 1 into two tables ( named as table 1 and table 2 in the revised version of the manuscript) and we have deleted lines 152-159 as suggested to avoid duplication of the information already presented in Table 1. We have briefly described the results of Table 2 in the Subdivision 3.2 of revised manuscript version as follows:

Almost all parents who intended to vaccine their children were aware of the vaccine’s approval for children and would vaccinate their children if the pediatrician recommended it, while both shares were lower among non-intenders (approval knowledge: 96.7% versus 88.9%, p=0.001; vaccinate if pediatrician recommended: 99.6% versus 16.6%, p<0.001)(Table 2). Attitudinal differences between intenders and non-intenders were also observed (Table 2). Parents who intended to vaccinate their children against COVID-19 exhibited disproportionately higher reliance on pediatricians (40.0% versus 13.6%, p<0.001) or other healthcare professionals (48.8% versus 28.6%, p<0.001) as their main sources to decide about their children’s vaccination, and they also reported higher support for mandating vaccination both for adults (64.2% versus 15.1%, p<0.001) and children (43.7% versus 0.0%, p<0.001). Similarly, most intenders reported a lot or very much trust in official healthcare guidelines compared to non-intenders (84.6% versus 29.1%, p<0.001) and an increase in their trust in the state and the healthcare system during the pandemic relatively to non-intenders (27.1% versus 2.5%. p<0.001). Flu vaccination was also higher in children of parents who intended to vaccinate them against COVID-19 compared to those who did not (31.3% versus 14.6%, p<0.001).

  1. Table 2, line 186, has unclear designations. It is not clear whether N=199 is the sum of the following numbers. The designation n=199 should be in the table's title, not in the first line. In the table, in addition to percentages, there should be numbers. 

Following your advice we have now amended previously named as Table 2 and now named as Table 3 by adding the total number in the title of Table 3 as well as numbers in the columns.

  1. In subsection 3.3, line 212, I propose to separate, as in the case of parents: Table 4 into two: characteristics of the respondents and the second table from the question: Have you been diagnosed...

We have now divided the Table into two new Tables named now Table 5 and Table 6

  1. Please consider bolding the questions in the tables - this will make distinguishing the question from the answer easier. 

We have now bolded the questions in the tables following your advice

  1. The results regarding pediatricians are treated very gloomily, perhaps because this is a small group; nevertheless, it would be possible to see, for example, how pediatricians who have children and pediatricians who do not have children relate to some of the questions and how those who agree with approving vaccination for children and those who disagree with it respond. 

This is indeed a very interesting comment. Due to the small number of the participating pediatricians as rightly noted we did not make any conclusions based on these findings

  1. The discussion needs to be improved by referring to more literature and comparing other studies. Many interesting papers have been published on parents and COVID-19 vaccination and pediatricians and COVID-19 vaccination. 

Indeed the childhood COVID-19 vaccination as well as parental and pediatricians` views in the field is an area with many papers published recently.

Following your comment we have expanded the discussion by adding more data on recent studies conducted worldwide and added the following references:

  • Wang L, Wen W, Chen C, et al.. Explore the attitudes of children and adolescent parents towards the vaccination of COVID-19 in China. Ital J Pediatr. 2022 Jul 23;48(1):122. doi: 10.1186/s13052-022-01321-7.
  • Ruiz JB, Bell RA. Parental COVID-19 Vaccine Hesitancy in the United States. Public Health Rep. 2022 Aug 2:333549221114346. doi: 10.1177/00333549221114346.
  • Nguyen KH, Nguyen K, Mansfield K, et al. Child and adolescent COVID-19 vaccination status and reasons for non-vaccination by parental vaccination status. Public Health. 2022 Jun 13;209:82-89. doi: 10.1016/j.puhe.2022.06.002

  1. Should the paragraph from line 278 be considered a Limitation of the Study? If so, it should be noted. 

We have added the following line in the paragraph after the strengths and before starting mentioning the limitations: We acknowledge that our study as a pragmatic one is not without limitations.

  1. I would also suggest highlighting the strengths of the study. E.g., I disagree with the sentence lines: 278-281 are strengths, and in my opinion, one might be tempted, with some caution, to try to transfer the results to countries in the region or the EU. 

This is indeed a very interesting comment that we have incorporated in the revised version of the submitted manuscript as follows: This study may differ from other similar studies, since our sample was selected at random from national phone-number databank. It includes parents from all the regions of Greece which is one of the major study strengths. Our results might be generalizable with caution to other countries in the region or within Europe with similar cultural and healthcare system-related factors

  1. One last technical note: Author Contributions, Informed Consent Statement, and Conflicts of Interest without quotation marks.

The quotation marks have now been removed